# Panoramic Ultrasound Examination of Posterior Neck Extensors in Healthy Subjects: Intra-Examiner Reliability Study

**DOI:** 10.3390/diagnostics10100740

**Published:** 2020-09-24

**Authors:** Juan Antonio Valera-Calero, Gracia María Gallego-Sendarrubias, César Fernández-de-las-Peñas, Joshua A. Cleland, Ricardo Ortega-Santiago, José Luis Arias-Buría

**Affiliations:** 1Department of Physical Therapy, Universidad Camilo José Cela, Villanueva de la Cañada, 28692 Madrid, Spain; gmgallego@ucjc.edu; 2Escuela Internacional de Doctorado, Universidad Rey Juan Carlos, 28933 Alcorcón, Spain; 3Department of Physical Therapy, Occupational Therapy, Rehabilitation and Physical Medicine, Universidad Rey Juan Carlos, 28922 Alcorcón, Spain; cesar.fernandez@urjc.es (C.F.-d.-l.-P.); ricardo.ortega@urjc.es (R.O.-S.); joseluis.arias@urjc.es (J.L.A.-B.); 4Cátedra Institucional en Docencia, Clínica e Investigación en Fisioterapia: Terapia Manual, Punción Seca y Ejercicio Terapéutico, Universidad Rey Juan Carlos, 28922 Madrid, Spain; 5Doctor of Physical Therapy Program, Department of Public Health and Community Medicine, Tufts University School of Medicine, Boston, MA 02155, USA; joshua.cleland@tufts.edu

**Keywords:** ultrasonography, panoramic ultrasound, neck muscles, reproducibility of results

## Abstract

Previous studies analyzing morphometry of posterior cervical muscles with ultrasound (US) imaging have mainly used Brightness mode (B-mode). Our aim was to investigate the intra-examiner reliability of panoramic US imaging for assessing posterior-lateral cervical muscle layers. Panoramic US images of the neck at C4/C5 level were acquired in 25 asymptomatic subjects (40% women; mean age: 24 years) by an experienced assessor. The cross-sectional area (CSA) of the upper trapezius, splenius, semispinalis, multifidi, rotators, and levator scapulae was measured from panoramic US scans on two separate days. Intra-class correlation coefficients (ICC3,1), standard error of measurement (SEM), minimal detectable change (MDC) and mean, absolute and percent errors were calculated. In general, intra-examiner reliability was excellent with ICC3,1 ranging from 0.978 (trapezius) to 0.993 (semispinalis). The SEM ranged from 0.02 (multifidus) to 0.07 (semispinalis/levator), whereas the MDC ranged from 0.05 (cervical multifidus) to 0.19 (levator/semispinalis). Absolute error was lower than 0.11 cm^2^ (levator/semispinalis). No differences between males and females were found. This study found that intra-examiner/rater reliability of panoramic US imaging was excellent for assessing the CSA of the posterior-lateral neck extensor muscles in asymptomatic subjects. The current findings suggest that panoramic US may be a reliable technique for examining the size of the cervical extensor muscles in both males and females.

## 1. Introduction

Ultrasound imaging (US) is an easy, safe, low-cost method for evaluating the musculoskeletal system. Brightness mode (B-mode) is generally used for assessing muscular echogenicity as indicator of muscle quality [1] and morphology measurements, e.g., thickness, depth, width, shape, or the cross-sectional area (CSA) [2]. Some studies have used B-mode to examine posterior neck muscle morphology in patients with neck pain [3]. The rationale for investigating muscle morphology is that neck pain induces changes in the posterior neck muscles, which could contribute to the chronification of pain symptoms. Although evidence supports the reduction of the CSA in the cervical multifidi in individuals with neck pain, the systematic review by DePauw et al. [4] concluded that more studies are needed as results from previous studies are inconsistent.

One limitation of B-mode is that the maximum structures size available for the assessment is limited to one screenshot, even if a convex probe is used. In fact, measuring the CSA of the posterior neck muscles is limited with B-mode. Consequently, extended areas cannot be acquired using B-mode. The development of US technology now permits a better acquisition method by including a quick, large and automatic construction of 2D cross-sectional images called panoramic view US (also called extended field-of-view). The panoramic US image consists of a set of images taken that are processed together to obtain a single image. All images in the set are registered, and they are stitched according to their transformations. Finally, the stitched image, i.e., the panoramic image, is obtained [5]. Panoramic US imaging has been found to be as valid as magnetic resonance imaging (MRI) for assessing muscle morphology, e.g., the CSA, within the quadriceps muscle [6,7]. In addition, the intra- and inter-examiner reliability of panoramic US imaging of muscle volume has been found to be excellent for assessing the CSA in the quadriceps muscle [8], gastrocnemius [9], hamstrings [10], and biceps brachii [11].

The posterior neck musculature consists of four different layers, from superficial to deep (at the C4 level), upper trapezius, splenius capitis and cervicis, semispinalis capitis and cervicis, and cervical multifidus and rotators [12]. Since posterior neck extensor muscles are important for the treatment of patients with neck pain [13], technical improvements in US imaging could permit better morphological assessment of cervical muscles. In such a scenario, panoramic US seems to be a promising method to assess muscles’ CSAs that cannot be measured using regular B-mode, which has not been previously used for assessing muscle morphology of the posterior neck extensors. An important step before clinicians can routinely use panoramic US view for investigating cervical spine muscle composition is to determine the reliability of the method. Previous studies using B-mode reported appropriate reliability for evaluating posterior neck muscle morphology, i.e., the CSA, mostly the cervical multifidi [14,15,16]. No study has investigated the reliability of panoramic US in the neck. Therefore, the objective of the current study was to determine the intra-examiner/rater reliability of panoramic US imaging for assessing the CSA of the posterior-lateral layers of the neck: 1, upper trapezius; 2, splenius (capitis/cervicis); 3, semispinalis (capitis/cervicis); 4, rotators; 5, cervical multifidi; and, 6, levator scapulae, as these muscles can be involved in neck pain symptoms.

## 2. Methods

### 2.1. Participants

Healthy volunteers without symptoms of neck pain were recruited by local announcements in Madrid (Spain) between October 2019 and January 2020. Volunteers were included if reported absence of neck pain the previous year and were between 18 and 45 years old. They were excluded if they presented with a previous history of whiplash; pharmacological treatment affecting muscle tone or pain perception; prior neck surgery; neuropathic conditions, e.g., radiculopathy or myelopathy; radiologic findings, e.g., degenerative findings; and any comorbid medical condition, e.g., tumor or fracture. This study was approved by the Institutional Review Ethical Committee of Universidad Rey Juan Carlos (URJC 3001201801618, 12 March 2018). Prior to their participation in the study, all participants were requested to read and sign the written informed consent.

### 2.2. Imaging Acquisition

All images were acquired with an Alpinion eCube8 i8 (Alpinion Medical Systems Co., Ltd., Gyeonggi-do, Korea) ultrasound equipment with a 3–12 MHz linear probe (E8-PB-L3-12T) by an examiner with more than 10 years of experience in musculoskeletal US imaging. The imaging acquisition procedure was conducted as recently described by Valera-Calero et al. [17]. Participants were prone with their head in a neutral position, their shoulders in 90° abduction, and their elbows flexed to 90°. All measurements were conducted at the C4/C5 level since this has exhibited less measurement error [16].

Equipment settings, including gain (55 dB), dynamic range (85), brightness (17), depth (4 cm), and frequency (12 MHz), were optimized for image quality using the musculoskeletal mode prior to testing. First, B-mode was used for identification of the C4/C5 level, that is, when the most superficial point of the spinous tubercle cortical C4 surface and the most superficial point of C4/C5 zygapophyseal joint were simultaneous [17]. The panoramic US images were captured by sliding laterally from the spinous process of the C4 vertebra to the posterior layer of the sternocleidomastoid muscle. Care was taken to ensure that consistent minimal pressure was applied to the skin with the US probe to avoid any muscle compression. To ensure that the US probe was moved perpendicular to the skin and along a transverse plane, a line from the C4 spinous process to the posterior layer of the sternocleidomastoid muscle was drawn. The software generates a real-time panoramic cross-sectional image where all posterior-lateral cervical muscle layers can be seen: upper trapezius, splenius, semispinalis, multifidi, rotators and levator scapulae (Figure 1A,B).

The CSA of each posterior cervical muscle layer, i.e., upper trapezius, splenius, semispinalis, multifidus, rotators and levator scapulae, was measured from panoramic US scans on two days 3 weeks apart in a randomized order. Every image was coded using alphanumerical codes for blinding the assessor to his previous assessment. The order of assessment was numerically randomized with computer software between participants.

### 2.3. Imaging Analysis

Once the images were captured, they were transferred to panoramic view offline DICOM Image-J software v.1.42 (National Institutes of Health, Bethesda, MD, USA) for calculating the CSA of the posterior neck muscle layers by using on-screen calipers traced around the following contours (Figure 1C):Upper trapezius—A, medial limit: internal echogenic trapezius fascia; B, superior limit: internal echogenic fascia between trapezius and skin; C, inferior limit: internal echogenic fascia between trapezius and splenius muscles.Splenius layer (capitis/cervicis muscles)—A, medial limit: internal echogenic fascia of splenius muscle; B, superior limit: internal fascia between splenius and trapezius medially and the skin laterally: C, inferior limit: internal fascia between splenius and semispinalis; D, lateral limit: internal fascia between splenius and sternocleidomastoid.Semispinalis layer (capitis/cervicis muscles)—A, medial limit: internal echogenic fascia of semispinalis muscle; B, superior limit: internal echogenic fascia between semispinalis and splenius muscle; C, inferior limit: internal echogenic fascia between semispinalis and cervical multifidus, rotators and transverse process; D, lateral limit: internal echogenic fascia between semispinalis and the vasculo-nervous bundle.Cervical multifidus—A, medial limit: echogenic spinous process; B, superior limit: echogenic fascia between cervical multifidus and semispinalis; C, inferior limit: internal echogenic fascia between multifidus and rotator muscles (deep to cervical multifidus).Rotators—A, inferior/medial limit: internal echogenic fascia over the cervical vertebra; B, superior limit: internal fascia between rotators and cervical multifidus; C, lateral limit: internal fascia between rotators and semispinalis.Levator scapulae—A, medial limit: internal echogenic fascia between levator scapulae and splenius/semispinalis layer; B, inferior limit: internal fascia between levator scapulae and medium scalene; C, lateral limit: internal fascia between levator scapulae and sternocleidomastoid muscle; D, superior limit: internal echogenic fascia between levator scapulae and skin.

### 2.4. Statistical Analysis

Before all analyses, each image was scaled from pixels to centimeters using the straight-line function in ImageJ. Statistical analyses were performed using the SPSS V.21 software for Mac OS (Armonk, NY, USA). Normal distribution of the data was verified using the Shapiro–Wilk test. Intra-examiner reliability for Panoramic US View was estimated using 2-way mixed-model, consistency-type intraclass correlation coefficient (ICC3,1). Intra-rater reliability was classified as fair (ICC < 0.50), moderate (0.5 < ICC < 0.75), good (0.75 < ICC < 0.9) or excellent (0.9 < ICC) [18]. Standard error of measurement (SEM = standard deviation * √1-ICC), minimal detectable change (MDC = 1.96 * SEM * √2), and the percent error (PE% = between-measurements absolute error/mean error of the measurements * 100) were also calculated.

## 3. Results

From a total of 43 subjects responding to the announcement, 18 were excluded due to a previous history of whiplash (*n* = 7) and history of chronic neck pain symptoms (*n* = 11). Twenty-five asymptomatic subjects (60% male) were finally included, obtaining 50 images (*n* = 2 per subject). Participants’ socio-demographic characteristics are described in Table 1.

Table 2 shows intra-examiner reliability of panoramic US for the posterior neck muscles. In general, intra-examiner reliability was excellent, ranging from 0.978 (upper trapezius) to 0.993 (semispinalis). The SEM ranged from 0.02 (multifidus) to 0.07 cm (levator/semispinalis), whereas the MDC ranged from 0.05 (multifidus) to 0.19 cm (levator/semispinalis).

The absolute error of measurement was less than 0.11 cm^2^ (semispinalis). No significant differences existed between males and females (Table 3).

## 4. Discussion

This is the first study to assess intra-examiner reliability of panoramic US imaging calculation of posterior neck muscle morphology. In general, intra-examiner reliability of panoramic US imaging was excellent for assessing the CSA of the posterior neck extensors, including the levator scapulae, in asymptomatic subjects showing small error values. Our findings suggest that panoramic US may be a reliable technique for examining the size of neck extensor muscles.

Previous studies have investigated reliability of panoramic US in examining large muscles of the lower or upper extremities [8,9,10,11]. Interestingly, the reliability values obtained in our study were similar to those previously observed for the quadriceps (ICC 0.963–0.991) [8], gastrocnemius (ICC 0.914) [9], hamstrings (ICC 0.715–0.984) [10] and biceps brachii (ICC 0.78–0.99) [11]. Panoramic US imaging has also exhibited excellent agreement with MRI for evaluating muscle size and changes in area, i.e., hypertrophy or atrophy, but mostly in large muscles of the lower extremity [6,7]. Future studies should examine the agreement between panoramic US imaging and MRI on the cervical spine.

One important advancement of panoramic US imaging is the ability to obtain a total image of large muscles, such as those located in the extremities, in a single image or the ability of visualizing different muscles, such as those located in the posterior and lateral part of the neck, in a single image, which is not possible with B-mode US imaging [5]. This is the reason why most previous studies using B-mode imaging in patients with neck pain have mainly calculated the CSA of the cervical multifidus, since this is the only muscle which can be visualized with the static B-mode in one image. In fact, panoramic US has also enabled the evaluation of the CSA of the levator scapulae for the first time. 

The current study can assist with developing exam protocols using panoramic US imaging in patients with neck pain for both research and clinical practice. Visualization of the posterior-lateral neck musculature layers in a single US image could better define morphological changes in this population and could also be used for monitoring changes in these muscles after the application of exercise programs in a single image.

Finally, it is important to mention that this study has some limitations. First, we included asymptomatic subjects. We do not know if similar results would be observed in patients with neck pain where fatty infiltration could decrease reliability data. Second, we only assessed intra-, not inter-, examiner reliability. Third, our sample size was relatively small. Therefore, our results should not be considered as potential normative data of the CSA of posterior-lateral neck muscles. Fourth, reliability data were obtained by an experienced assessor, so we do not currently know if an unexperienced evaluator could obtain similar results with panoramic US imaging.

## 5. Conclusions

We found that panoramic ultrasound assessment of posterior and lateral cervical musculature layers at C4/C5 level is highly reliable for evaluating the CSA in asymptomatic people since imaging calculations showed excellent intra-examiner reliability. Reliability was similar in both males and females. Future studies should investigate if panoramic US imaging can be used for evaluating cervical musculature changes in patients with neck pain symptoms.

## Figures and Tables

**Figure 1 diagnostics-10-00740-f001:**
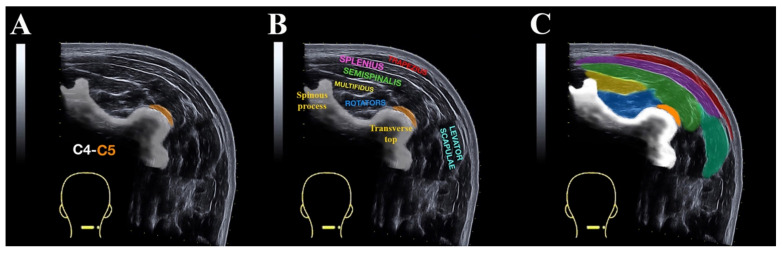
Panoramic ultrasound imaging at C4 to C5 level showing the superficial posterior neck muscles. (**A**) Image without labelling; (**B**) image with muscular names labelled; (**C**) the cross sectional area (CSA) assessment of the upper trapezius (red), splenius layer (pink), semispinalis layer (green), multifidi (yellow), rotator (blue), and levator scapulae (light green).

**Table 1 diagnostics-10-00740-t001:** Participants’ socio-demographic characteristics.

Characteristics	Male (*n* = 15)	Female (*n* = 10)	Total (*n* = 25)
Age (years)	22.5 ± 4.5	25.5 ± 6.0	23.8 ± 5.0
Height (meters) *	1.8 ± 0.05	1.7 ± 0.05	1.75 ± 0.1
Weight (kg) *	77.5 ± 7.0	56.8 ± 4.4	69.25 ± 12.0
BMI (kg/m^2^) *	24.0 ± 2.0	20.25 ± 1.9	22.5 ± 2.7

Values are expressed as mean ± SD; * significant differences between male and female (*p* < 0.05).

**Table 2 diagnostics-10-00740-t002:** Intra-examiner reliability of panoramic ultrasound imaging.

Reliability Estimates	CM	SR	SS	SP	LE	Tr
Mean (cm^2^)	1.06 ± 0.22	1.72 ± 0.43	2.65 ± 0.76	2.64 ± 0.70	3.17 ± 0.76	1.09 ± 0.41
Error (cm^2^)	−0.01 ± 0.04	0.03 ± 0.10	0.02 ± 0.12	0.01 ± 0.14	−0.01 ± 0.15	−0.00 ± 0.12
Absolute Error (cm^2^)	0.03 ± 0.03	0.08 ± 0.07	0.10 ± 0.07	0.11 ± 0.09	0.11 ± 0,10	0.08 ± 0.08
ICC_3,1_ (95% CI)	0.989 (0.981–0.994)	0.983 (0.971–0.991)	0.993 (0.988–0.996)	0.989 (0.981–0.994)	0.990 (0.983–0.994)	0.978 (0.962–0.988)
SEM	0.02	0.05	0.06	0.07	0.07	0.06
MDC	0.05	0.13	0.16	0.19	0.19	0.16
PE (%)	2.83	4.65	3.77	4.16	3.47	7.33

CI: confidence interval; ICC: intraclass correlation coefficient; MDC: minimum detectable changes; PE: percent error; SEM: standard error of measurement; Tr: upper trapezius; LE: levator scapulae; SP: splenius layer; SS: semispinalis layer; SR: short rotators; CM: cervical multifidus.

**Table 3 diagnostics-10-00740-t003:** Intra-examiner reliability of panoramic ultrasound imaging by Gender.

Reliability Estimates	CM	SR	SS	SP	LE	Tr
	**Males (*n* = 15)**					
Mean (cm^2^)	1.12 ± 0.22	1.85 ± 0.41	3.07 ± 0.65	2.95 ± 0.65	3.41 ± 0.73	1.31 ± 0.38
Error (cm^2^)	−0.01 ± 0.04	0.02 ± 0.12	0.03 ± 0.13	0.03 ± 0.12	−0.03 ± 0.12	0.00 ± 0.15
Absolute Error (cm^2^)	0.03 ± 0.02	0.09 ± 0.08	0.11 ± 0.08	0.09 ± 0.08	0.10 ± 0.10	0.11 ± 0.09
ICC_3,1_ (95% CI)	0.990 (0.979–0.995)	0.978 (0.954–0.990)	0.989 (0.977–0.995)	0.990 (0.979–0.995)	0.990 (0.979–0.995)	0.961 (0.918–0.981)
SEM	0.02	0.06	0.06	0.06	0.07	0.07
MDC	0.05	0.16	0.16	0.16	0.19	0.19
PE (%)	2.67	4.86	3.58	3.05	2.93	8.39
	**Females (*n* = 10)**					
Mean (cm^2^)	0.98 ± 0.20	1.53 ± 0.37	2.02 ± 0.39	2.16 ± 0.48	2.81 ± 0.67	0.76 ± 0.18
Error (cm^2^)	−0.01 ± 0.05	0.05 ± 0.08	0.00 ± 0.10	−0.01 ± 0.16	0.02 ± 0.15	−0.01 ± 0.05
Absolute Error (cm^2^)	0.03 ± 0.03	0.07 ± 0.06	0.08 ± 0.06	0.13 ± 0.09	0.12 ± 0.09	0.04 ± 0.04
ICC_3,1_ (95% CI)	0.984 (0.960–0.994)	0.987 (0.967–0.995)	0.981 (0.953–0.993)	0.969 (0.923–0.988)	0.987 (0.966–0.995)	0.975 (0.938–0.990)
SEM	0.02	0.04	0.05	0.08	0.07	0.02
MDC	0.05	0.11	0.13	0.22	0.19	0.05
PE (%)	3.06	4.57	3.96	6.01	4.27	5.26

CI: confidence interval; ICC: intraclass correlation coefficient; MDC: minimum detectable changes; PE: percent error; SEM: standard error of measurement; Tr: upper trapezius; LE: levator scapulae; SP: splenius layer; SS: semispinalis layer; SR: short rotators; CM: cervical multifidus.

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
