# Peer review of "Panoramic Ultrasound Examination of Posterior Neck Extensors in Healthy Subjects: Intra-Examiner Reliability Study"

_diagnostics, 2020, doi:10.3390/diagnostics10100740_

Round 1
Reviewer 1 Report
the article has an interesting topic but English and spelling errors should be reviewed.
references should be more recent.
introduction:
it must refer to the muscular ultrasound evaluation through echogenicity as a practice both in terms of characterization and diagnosis.
methods
why just used the line between c4-c5 and not other references and then calculate an average? since this location is not the muscular belly of all the studied muscles.
the study was not submitted to an ethics committee? if yes, the document number must be presented.
results
SEM could be presented by splots.
discussion
it should be more exhaustive in terms of comparing the results with the literature.
conclusion
it must be direct and present future ideas.
Author Response
We thank the reviewers for their comments, which have clarified several aspects of the manuscript. We would like to make the Editor know that we addressed all suggestions made by the reviewers and the assistant editor. All changes are highlighted to facilitate the review process.
Reviewer 1
The article has an interesting topic but English and spelling errors should be reviewed
Response: We would like to thank to the reviewer for the positive feedback. The English and spelling errors have been reviewed by one American native co-author.
References: Should be more recent
Response: We want to comment to the reviewer that we have cited the most relevant and recent studies focusing on the topic of this study.
Introduction: It must refer to the muscular ultrasound evaluation through echogenicity as a practice both in terms of characterization and diagnosis
Response: We have now included the echogenicity assessment in the muscular ultrasound evaluation (Lines 38-39)
Methods: Why just used the line between C4-C5 and not other references and then calculate an average? since this location is not the muscular belly of all the studied muscles.
Response: As reported in line 92.93, all measurements were conducted at that level since this has exhibited less measurement error.
The study was not submitted to an ethics committee? if yes, the document number must be presented.
Response: The Ethics Committee approval number is reported in line 83 (URJC….)
Results: SEM could be presented by splots.
Response: All data, including SEM are included in tables. We believe that a figure would not add to the info provided in the table.
Discussion: It should be more exhaustive in terms of comparing the results with the literature
Response: Since this is the first study assessing the reliability of panoramic ultrasound in the neck extensors, it was not possible to compare our results with previous studies. We have included all the available evidence in this topic.
Conclusion: It must be direct and present future ideas.
Response: We believe that the conclusion summarizes adequately the main findings according with the objectives of the study and future research lines are also reported.
We hope that the current version of the manuscript can get a positive review and can be accepted for publication in Diagnostics
Sincerely yours,
The authors
Reviewer 2 Report
There is many pathological conditions relating cervical muscles, from muscle diseases, i.e. various metabolic myopathy, localized dystrophy, to neuropathies, i.e. motor neuron disease and some of dropped head syndrome. I think this paranomic US may be important tool to examine the cause and check the effect of intervention.
Author Response
We thank the reviewers for their comments, which have clarified several aspects of the manuscript. We would like to make the Editor know that we addressed all suggestions made by the reviewers and the assistant editor. All changes are highlighted to facilitate the review process.
Reviewer 2
There is many pathological conditions relating cervical muscles, from muscle diseases, i.e. various metabolic myopathy, localized dystrophy, to neuropathies, i.e. motor neuron disease and some of dropped head syndrome. I think this paranomic US may be important tool to examine the cause and check the effect of intervention.
Response: We would like to thank to the reviewer for the positive feedback
We hope that the current version of the manuscript can get a positive review and can be accepted for publication in Diagnostics
Sincerely yours,
The authors